# Alterations in the Antioxidant Enzyme Activities in the Neurodevelopmental Rat Model of Schizophrenia Induced by Glutathione Deficiency during Early Postnatal Life

**DOI:** 10.3390/antiox9060538

**Published:** 2020-06-19

**Authors:** Magdalena Górny, Anna Bilska-Wilkosz, Małgorzata Iciek, Marta Hereta, Kinga Kamińska, Adrianna Kamińska, Grażyna Chwatko, Zofia Rogóż, Elżbieta Lorenc-Koci

**Affiliations:** 1The Chair of Medical Biochemistry, Jagiellonian University Medical College, 7 Kopernika Street, 31-034 Kraków, Poland; mbgorny@cyf-kr.edu.pl (M.G.); mbbilska@cyf-kr.edu.pl (A.B.-W.); miciek@cm-uj.krakow.pl (M.I.); 2Maj Institute of Pharmacology, Polish Academy of Sciences, 12 Smętna Street, 31-343 Kraków, Poland; hereta@if-pan.krakow.pl (M.H.); k.kamin@if-pan.krakow.pl (K.K.); rogoz@if-pan.krakow.pl (Z.R.); 3Department of Environmental Chemistry, University of Łódź, 163 Pomorska Street, 90-236 Łódź, Poland; adka367@interia.eu (A.K.); grazyna.chwatko@chemia.uni.lodz.pl (G.C.)

**Keywords:** neurodevelopmental model of schizophrenia, GSH deficiency, antioxidant enzymes activity, ROS level

## Abstract

The aim of the present study was to assess the effects of l-buthionine-(S,R)-sulfoximine (BSO), a glutathione (GSH) synthesis inhibitor, and GBR 12909, a dopamine reuptake inhibitor, administered alone or in combination to Sprague-Dawley rats during early postnatal development (p5–p16), on the levels of reactive oxygen species (ROS), lipid peroxidation (LP) and the activities of antioxidant enzymes: superoxide dismutase (SOD), catalase (CAT), glutathione peroxidase (GPx) and glutathione disulfide reductase (GR) in peripheral tissues (liver, kidney) and selected brain structures (prefrontal cortex, PFC; hippocampus, HIP; and striatum, STR) of 16-day-old rats. The studied parameters were analyzed with reference to the content of GSH and sulfur amino acids, methionine (Met) and cysteine (Cys) described in our previous study. This analysis showed that treatment with a BSO + GBR 12909 combination caused significant decreases in the lipid peroxidation levels in the PFC and HIP, in spite of there being no changes in ROS. The reduction of lipid peroxidation indicates a weakening of the oxidative power of the cells, and a shift in balance in favor of reducing processes. Such changes in cellular redox signaling in the PFC and HIP during early postnatal development may result in functional changes in adulthood.

## 1. Introduction

There is a growing body of evidence implicating oxidative stress mechanisms in the pathophysiology of schizophrenia [1,2,3,4,5]. The most central feature of oxidative stress is disruption of the redox states of thiol systems represented by three thiol/disulfide redox couples such as: (1) reduced glutathione (GSH) and its disulfide (GSSG), GSH/GSSG couple; (2) cysteine (Cys) and its disulfide, cystine (CySS), Cys/CySS couple; and (3) reduced and oxidized thioredoxins (Trx), Trx^red^/Trx^ox^ couple, which are normally involved in cell-to-cell signaling, macromolecular trafficking and physiological regulations [6]. Biological systems generate much more non-radical oxidants than free radicals, such as hydrogen peroxide (H_2_O_2_), peroxynitrite (ONOO^−^), aldehydes, quinones and disulfides [7]. All these oxidants significantly affect the regulation of the cellular redox state by modulating sulfhydryl (–SH) residues of Cys and thioeter groups of methionine (Met) located in the active sites of many proteins [7]. These redox molecules are usually susceptible to two-electron oxidants. Therefore, the adequate levels of thiol antioxidants, like GSH, Cys and Trx^red^, which control the functions of the thiol-containing proteins, are essential for maintaining the physiological redox status of cells.

For almost twenty years, the dysregulation of GSH synthesis has been postulated as an important factor in the pathophysiology of schizophrenia [8,9,10,11,12,13]. Disturbances in the GSH biosynthesis in some schizophrenia patients have been shown to be linked with polymorphisms in the genes encoding both catalytic and modifier subunits of γ-glutamate-cysteine ligase (GCL) [14,15,16], a pivotal enzyme in the two-step reaction of GSH biosynthesis. In schizophrenic patients with such gene modifications, the severity of negative symptoms correlated well with a decrease in GSH levels in the brain [17]. Furthermore, it was reported that the level of GSH was reduced in the medial prefrontal cortex (mPFC) and cerebrospinal fluid of drug-naive schizophrenic patients [9,10,12,13] as well as in the striatum (STR) and PFC of those treated earlier with antipsychotic drugs [8,11].

Similarly to other clinical studies, the occurrence of schizophrenia-like symptoms [18,19,20,21,22,23]—as well as numerous biochemical and morphological alterations, reminiscent of those observed in patients—have been described in experimental animals [24,25,26,27,28], in which the GSH deficit was induced by genetic manipulation in genes encoding GCL enzyme [29,30,31], or by specific compounds that reduce GSH concentration [32,33]. In our recently published study (Górny et al. [34]), the chronic administration of l-buthionine-(S,R)-sulfoximine (BSO), a specific GCL inhibitor, to newborn male Sprague-Dawley rats between the postnatal day p5 and p16, has been shown to induce the tissue-dependent decreases in GSH contents, paralleled by alterations in homeostasis of sulfur-containing amino acids (Met and Cys), when measured 4 h after the last dose. As a consequence of the changes in the contents of these thiols in peripheral tissues (liver, kidneys), and in certain brain structures (prefrontal cortex, PFC; hippocampus, HIP) during early postnatal development, schizophrenia-like behaviors appeared in adulthood (p90–p93), which were accompanied by changes in the global DNA methylation status in the studied brain structures [34]. However, to date, the activities of antioxidant enzymes, such as superoxide dismutase (SOD), catalase (CAT), glutathione peroxidase (GPx) and glutathione disulfide reductase (GR) that control the levels of reactive oxygen species (ROS), such as superoxide anion radical (O_2_^•−^) and hydrogen peroxide (H_2_O_2_), and are principal determinants of the redox state of cells, have not been studied in the rat model of schizophrenia induced by BSO treatment during the early postnatal life.

It is worth recalling that O_2_^•−^ and H_2_O_2_ are normal products of cellular aerobic metabolism that can play a dual role both as beneficial and harmful molecules [35,36]. The beneficial effect of low or moderate concentrations of these molecules, especially H_2_O_2_, termed “oxidative eustress” (physiological redox signaling), is associated with their activity as the second messengers, capable of regulating many intrinsic signaling pathways and promoting different physiological functions, from the innate immune response to neuronal development [37,38,39,40,41,42,43]. Harmful effects of these molecules defined as “oxidative distress” (pathologically disrupted redox signaling) [41,43], occur when their action leads to the formation of secondary ROS, such as hydroxyl radical (HO^•^) and ONOO^–^, which are highly reactive molecules capable of inflicting significant damage to important cellular proteins [36,44]. Considering the data presented above in the context of pathogenesis of schizophrenia, it seems that examining the effects of GSH deficiency in the early postnatal life, both on the activity of antioxidant enzymes (SOD, CAT, GPx, GR) and on ROS concentrations, may be of vital importance for defining the role of oxidative stress in the development of this disease.

Therefore, to clarify this issue, and to find a relationship between changes in the activity of antioxidant enzymes (SOD, CAT, GPx, GR), and the contents of GSH and sulfur amino acids (Cys, Met), described in our recently published study [34], we measured the activities of these enzymes and the levels of ROS as well as lipid peroxidation expressed as malondialdehyde (MDA) level, in the same groups of 16 day-old Sprague-Dawley rats administered chronically with BSO and the compound GBR 12909, which is an inhibitor of dopamine transporter (DAT), alone or in combination. The rationale for the administration of the DAT inhibitor was to check whether an increase in the extracellular dopamine (DA) level may exert some specific additional effect on both the activities of these enzymes and ROS levels. As in our earlier study [34], these model compounds (BSO, GBR 12909) were administered chronically between the postnatal day 5 and 16. Similarly, the activities of antioxidant enzymes were determined in the tissue homogenates of peripheral organs (liver and kidney) and the brain structures involved in the expression of schizophrenia-like symptoms (PFC and HIP), 4 h after the last BSO, and/or GBR 12909 doses. In the previous study [34], the effects of BSO and GBR 12909 on GSH and sulfur amino acid levels in the STR were not analyzed. However, because this structure has the highest DA content in the brain and plays a significant role in the appearance of positive symptoms of schizophrenia, the present studies on the activity of antioxidant enzymes and ROS levels have been supplemented with data on the concentration of GSH, Cys and Met in this structure. We hope that the analysis of the obtained data will contribute to an explanation of the role of oxidative stress in the development of schizophrenia.

## 2. Materials and Methods

The experiments were performed in compliance with the Act on the Protection of Animals Used for Scientific or Educational Purposes of 21 January 2005 reapproved on 15 January 2015 (published in Journal of Laws no 23/2015 item 266, Poland), and according to the Directive of the European Parliament and of the Council of Europe 2010/63/EU of 22 September 2010 on the protection of animals used for scientific purposes. They also obtained an approval of the Local Ethics Committee at the Institute of Pharmacology, Polish Academy of Sciences (permission no 3/2018 of 11 January 2018). The experimenters have made every effort, to reduce the number of animals used and their suffering.

### 2.1. Animals and Treatment

To create the neurodevelopmental model of schizophrenia, pregnant Sprague-Dawley females were delivered to our laboratory by the Charles River Company (Sulzfeld, Germany). They were kept in individual cages, with free access to standard laboratory food and tap water under standard laboratory conditions i.e., at room temperature (22 °C) and under an artificial light/dark cycle (12/12 h). On the day of delivery, the sex of pups was determined. Only male pups which were left with their mothers were treated on the postnatal days p5–p16, with the selective inhibitor of GCL, compound l-buthionine-(S,R)-sulfoximine (BSO) and the DA reuptake inhibitor GBR 12909, alone or in combination. BSO at a dose of 3.8 mmol/kg s.c., was administered once daily, while GBR 12909 at a dose of 5 mg/kg s.c., every second day. Both model compounds were dissolved in 0.9% NaCl. Under the combined treatment regimen, BSO administration proceeded GBR 12909 injection. Control rats instead of the model compounds received a 0.9% NaCl once daily. Every day, pups were weighed and the injected volumes of the studied model compounds were adjusted accordingly with the actual body weight. On postnatal day p16, the rats were killed by decapitation 4 h after the last doses of model compounds, and their liver, kidney and selected brain structures (PFC, HIP, STR) were dissected, immediately frozen on dry ice and stored at −80 °C until further biochemical analysis.

### 2.2. Chemicals and Reagents

1-[2-[Bis(4-fluorophenyl)methoxy]ethyl]-4-(3-phenylpropyl)piperazine dihydrochloride (GBR 12909) was received from Abcam Biochemicals (Cambridge, UK). l-Buthionine-(S,R)-sulfoximine (BSO), 7-dichlorofluorescein (DCF), 2,7-dichlorofluorescein diacetate (DCFH-DA), dithiothreitol (DTT), glutathione (GSH), hydrogen peroxide (H_2_O_2_), methionine (Met), N-acetyl-cysteine (NAC), O-phthaldialdehyde (OPA), *p*-phenylenediamine, sodium hydroxide (NAOH), sodium hydrosulfide (NaHS), 1,1,3,3-tetraethoxypropane (TEP), thiobarbituric acid (TBA), trichloroacetic acid (TCA) and tris-(2-carboxyethyl)phosphine (TCEP) were purchased from Sigma-Aldrich Chemical Company (St. Louis, MO, USA). 2-Chloro-1-methylquinolinium tetrafluoroborate (CMQT) was prepared according to the procedure described by Bald and Głowacki [45] in the Department of Environmental Chemistry, University of Łódź (Łódź, Poland). Catalase Assay Kit (Item No 707002), Glutathione Peroxidase Assay Kit (Item No 703102), Glutathione Reductase Assay Kit (Item No 703202) and Superoxide Dismutase Assay Kit (Item No 706002) were received from the Cayman Chemical Company (Ann Arbor, MI, USA). HPLC-grade acetonitrile (ACN) was from J.T. Baker (Deventer, The Netherlands), while perchloric acid (PCA) was obtained from Merck (Darmstadt, Germany). Folin-Ciocalteu phenol reagent, ferric chloride (FeCl_3_), hydrochloric acid (HCl), sodium dihydrogen phosphate dihydrate (NaH_2_PO_4_ × 2H_2_O), sodium hydrogen phosphate heptahydrate, (Na_2_HPO_4_ × 7H_2_O) were purchased from the Polish Chemical Reagent Company (P.O.Ch, Gliwice, Poland).

### 2.3. Preparation of Tissue Homogenates

The frozen tissues were weighed and immediately homogenized at an operating speed of 6000 rpm using an IKA-ULTRA-TURRAX T10 homogenizer (IKA Poland sp. z o.o company, Warsaw, Poland). For the determination of CAT activity in the liver and kidney, the samples were homogenized in 0.1 M phosphate buffer, pH 7.4 (1 g of tissue in 4 mL buffer) for 1 min, while all tissue samples for other biochemical analysis were homogenized in buffers and at the ratio (*w/v*), according to the appropriate kit procedure. Homogenates were next used for biochemical assays.

### 2.4. Determination of Total Superoxide Dismutase (SOD) Activity

SOD activity was measured by colorimetric assay using Cayman’s kit (Cayman Chemical Company, Ann Arbor, MI, USA). Superoxide radical anion (O_2_^•−^), a SOD substrate, was generated by xantine oxidase and hypoxantine. SOD solution with known activity was used as a standard (0–0.05 U/mL). SOD activity was expressed in U/mg of protein (the amount of enzyme needed for 50% dismutation of O_2_^•−^ radical).

### 2.5. Determination of Catalase (CAT) Activity

CAT activity in the liver and kidney was determined according to the method described by Aebi [46]. CAT catalyzes H_2_O_2_ reduction which causes a decrease in absorbance at a wavelength of 240 nm.

Briefly, 50 μL of liver or kidney homogenate (diluted 50-fold) was added to 650 μL of 50 mM phosphate buffer (pH 7.0), and then the reaction was started by addition of 300 μL of 54 mM H_2_O_2_. After a minute, the decline in absorbance was measured. The blank contained 50 mM phosphate buffer, pH 7.0 and 300 μL of 54 mM H_2_O_2_. One unit of CAT activity corresponded to the amount of the enzyme that degraded 1 μmol H_2_O_2_ per minute at 25 °C. The activity of CAT in the liver and kidney was expressed in μmol/mg of protein/min.

CAT activity in brain structures (due to the small amount of biological material) was assayed using CAT Assay Kit (Cayman Chemical Company, Ann Arbor, MI, USA). The method involves spectrophotometric measurement of formaldehyde formed in the reaction of methanol with H_2_O_2_, catalyzed by CAT. The formaldehyde concentrations were read from a calibration curve prepared for the formaldehyde standards (0–75 μM). One unit of CAT activity was defined as the amount of the enzyme that caused the formation of 1 nmol of formaldehyde per minute at 25 °C. The activity of CAT was expressed in nmol/mg of protein/min.

### 2.6. Determination of Glutathione Peroxidase (GPx) Activity

GPx activity was assayed using the Cayman’s Glutathione Peroxidase Assay Kit (Cayman Chemical Company, Ann Arbor, MI, USA). The oxidized glutathione (GSSG) formation reaction catalyzed by GPx is coupled with the glutathione disulfide reductase (GR) reaction, which regenerates GSH at the expense of nicotinamide adenine dinucleotide phosphate (NADPH) oxidation. NADPH oxidation results in a decrease in absorbance at 340 nm, which can be easily measured spectrophotometrically.

The measure of GPx activity in the tested samples is the difference in the rate of absorbance change (∆A_340_/min) measured for the sample containing the enzyme, and in the rate of decrease in absorbance (∆A_340_/min) for the control sample. The rate of decrease in the A_340_ is directly proportional to the GPx activity in the sample. GPx activity was expressed in nmol/mg of protein/min (nmol of NADPH oxidized to NADP^+^ by the enzyme during 1 min per mg of protein).

### 2.7. Determination of Glutathione Disulfide Reductase (GR) Activity

Determination of GR activity was carried out using the Glutathione Reductase Assay Kit (Cayman Chemical Company, Ann Arbor, MI, USA) by measuring the rate of NADPH oxidation to NADP^+^. GR catalyzes the reduction of GSSG to its reduced form GSH, which is accompanied by NADPH oxidation. The oxidation of NADPH causes a decrease in absorbance at 340 nm, and is directly proportional to the GR activity in the sample. GR activity was expressed in nmol/mg protein/min (nmol of NADPH oxidized to NADP^+^ by the enzyme during 1 min per mg of protein).

### 2.8. Determination of Reactive Oxygen Species (ROS) Level

The determination of ROS was based on the method of Bondy and Guo [47]. In this designation 2,7-dichlorofluorescin diacetate (DCFH-DA) is hydrolyzed in homogenates to 2,7-dichlorohydrofluorescein (DCFH), and then oxidized to fluorescent 2,7-dichlorofluorescein (DCF) by ROS.

Briefly, 10 μL of 1.25 M DCFH-DA dissolved in ethanol and 10 μL of homogenate were added to 990 μL of 0.1M phosphate buffer (pH 7.4). Next, the mixture was incubated at 37 °C for 30 min, protected from light. Fluorescence of the product was measured at wavelengths: A_ex_ = 488 nm and A_em_ = 525 nm. ROS were evaluated from a standard curve with 10 μM DCF, and were presented in nmoles of DCF per g tissue.

### 2.9. Determination of the Concentration of MDA

The method used by us was based on the determination of MDA concentration as a product of lipid peroxidation, in the reaction with thiobarbituric acid (TBA), in a spectrophotometric assay. 1,1,3,3-Tetraethoxypropane (TEP) was used as a standard [48].

Briefly, 250 μL of 15% TCA and 250 μL of 0.37% TBA were added to 125 μL of homogenate and the samples were vortexed and incubated at 100 °C for 10 min. Then, the mixtures were centrifuged at 12,000× *g* for 5 min at 4 °C. The collected supernatant was again centrifuged, and absorbance was measured at a wavelength λ = 535 nm. Concentration of MDA in homogenates was calculated using a standard curve prepared with 25 µM TEP and was expressed in nmoles of MDA per g tissue.

### 2.10. Determination of Total GSH and Cys Levels in the Striatum

The high-performance liquid chromatography (HPLC) with ultraviolet (UV) detection was used for determination of total Cys and GSH levels in the striatal samples. The HPLV-UV method is based on the reduction disulfide bonds by tris(2-carboxyethyl) phosphine (TCEP) prior to precolumn derivatization with 2-chloro-1-methylquinolinium tetrafluoroborate (CMQT) [49,50].

Briefly, to 100 µL of the striatal tissue homogenate 7.5 µL of 0.25 M TCEP solution was added, vortexed and put aside for 15 min at room temperature, followed by addition of 10 µL of 0.1 M CMQT. After 5 min, the mixture was acidified with 15 µL of 3 M perchloric acid (PCA) followed by centrifugation (12,000 rpm, 10 min, 10 °C). The supernatant of samples were transferred into an autosampler vial and aliquot of 10 µL was injected into the HPLC system (1220 Infinity LC from Agilent, Waldbronn, Germany), equipped with a diode-array detector and commanded by OpenLAB CDS ChemStation Edition software. Each sample was analyzed in duplicate with the used of Zorbax SB-C18 column (150mm × 4.6 mm, 5 µm), and the mobile phase consisted of acetonitrile (ACN), and 0.1 M TCA adjusted by 1M NaOH to pH = 1.6. The detector wavelength of 355 nm, temperature of 25 °C, flow-rate of 1 mL/min and gradient elution were applied. The elution profile was as follows: 0–3.5 min, 11–25% (ACN); 3.5–5.5 min, 25–40% (ACN); 5.5–9 min 40–11% (ACN).

The retention times and diode-array spectra taken at real time of analysis were used for the identification of Cys and GSH peaks. Cys and GSH concentrations were established using calibration curves prepared in the ranges of 0–30 and 0–300 nmol/mL, respectively. The content of Cys and GSH in the striatum tissue were expressed in nmoles Cys or GSH per g tissue.

### 2.11. Determination of Met Level in the Striatum

The HPLC with fluorescence (FL) detection was used for determination of Met level in the striatum tissue. The modified HPLC-FL method [51] is used for the determination of primary amines (Met) and also thiols, and is based on an on-column derivatization with *o*-phthaldialdehyde (OPA).

Briefly, 100 µL of tissue homogenate 14 µL of 0.25 M TCEP solution was added, vortexed and kept for 10 min at room temperature. The mixture was treated with 30 µL of 0.5 M *N*-acetyl-cysteine solution and 10 µL of 3 M PCA, vortexed and centrifuged (12,000 rpm, 10 min, 10 °C). The supernatant was transferred into autosampler vial and injected (5 µL) into the HPLC system (1100 Series from Hewlett-Packard, Waldbronn, Germany), equipped with the column PRP-1 Hamilton (150 nm × 4.6 nm, 5 µm), an FL detector (1260 Series from Agilent Technologies, Waldbronn, Germany) and commanded by HP ChemStation software. Each sample was analyzed in duplicate with the used of the mobile phase consisted of ACN and 0.01 M OPA in 0.1 M NaOH pumped at 1 mL/min in the gradient mode as follow: 0–8 min, 14–25% (ACN); 8–12 min, 25% (ACN), 12–14 min, 25–14% (ACN). The FL detector was set at 348 nm and 438 nm for excitation and emission wavelengths, respectively. Met peak identification was based on comparison of retention time taken at real time of analysis with data obtained for standard compounds. Met level was read from the calibration curves prepared in the range of 0–20 nmol/mL, and was finally expressed in nmoles Met per g tissue.

### 2.12. Determination of Bound Sulfane Sulfur

Bound sulfane sulfur was determined by the modified method of Ogasawara et al. [52]. The bound sulfur is released during reduction by dithiothreitol (DTT) and the sulfide ions formed in this reaction react with *p*-phenylenediamine in the presence of ferric chloride (FeCl_3_), yielding a fluorescence dye thionine. The product is measured fluorometrically.

In short, 250 μL of 20 mM DTT and 50 μL of 50 mM borate buffer (pH = 9.2) were added to 125 μL of homogenate and the mixture was kept at 37 °C for 10 min. Next, 10 µL of 0.1 M NaOH, 400 µL of 12.5 mM *p*-phenylenediamine and 100 µL of 40 mM FeCl_3_ in 6 M HCl were added, and this reaction mixture was again incubated for 10 min at 37 °C. Next, the samples were centrifuged at 13,400× *g* for 5 min at room temperature. The fluorescence of the mixture was measured at wavelengths: A_ex_ = 600 nm and A_em_ = 623 nm. The bound sulfane sulfur level was calculated using a standard curve prepared with NaHS (10–100 µM) and was expressed in nmoles of NaSH per g tissue.

### 2.13. Determination of Protein Content

The protein concentration in tissue homogenates was determined using the method of Lowry et al. [53]. This method uses two separate reactions: the reaction of aromatic amino acids with the Folin-Ciocalteau reagent and the reaction of peptide bonds with cupric ions in alkaline environment (biuret reaction).

### 2.14. Statistics

The obtained data were analyzed using a two-way analysis of variance (ANOVA) followed (if significant) by Bonferroni test for post-hoc comparisons.

## 3. Results

### 3.1. The Effects of Chronic Treatment with BSO and GBR 12909 on the Enzymatic Activities of Superoxide Dismutase (SOD), Catalase (CAT), Glutathione Peroxidase (GPx) and Glutathione Disulfide Reductase (GR) in the Liver and Kidney of 16-Day-Old Rats

Figure 1 shows the activity of antioxidant enzymes (SOD, CAT, GPx, GR) in the liver and kidneys of male Sprague-Dawley pups, measured in tissue homogenates, 4 h after the last doses of BSO and GBR 12909 administered alone or in combination.

With respect to SOD activity in the liver, a two-way ANOVA showed no treatment effects of both BSO (F_(1,28)_ = 1.513, NS—non significant) and GBR 12909 (F_(1,28)_ = 1.490, NS), as well as no interaction between these two model substances (F_(1,28)_ = 1.088, NS) (Figure 1A). In line with the above effects, no post-hoc comparison of the studied groups was performed. The same analysis carried out for SOD activity in the rat kidneys also revealed a lack of treatment effects of both BSO (F_(1,28)_ = 1.397, NS) and GBR 12909 (F_(1,28)_ = 1.505, NS), but a significant interaction between these two model substances (F_(1,28)_ = 13.471, *p* < 0.001) was observed (Figure 1B). Comparisons of SOD activity between the studied rat groups showed that both BSO and GBR 12909, given alone, significantly increased the activity of this enzyme by 181.5% and 183.3% of the control value, respectively (Figure 1B).

As to the CAT activity in the liver of rats treated chronically with the model compounds, a two-way ANOVA demonstrated a lack of BSO treatment effect (F_(1,28)_ = 0.085, NS), but a significant overall GBR 12909 treatment effect (F_(1,28)_ = 7.702, *p* < 0.01), and no interaction between BSO and GBR 12909 (F_(1,28)_ = 2.423, NS). Post hoc comparisons showed that only GBR 12909 significantly decreased hepatic activity of this enzyme (Figure 1C). An analogous analysis carried out for CAT activity in the kidneys did not reveal any significant treatment effects of both BSO (F_(1,28)_ = 0.680, NS) and GBR 12909 (F_(1,28)_ = 0.611, NS), but only a significant interaction between these two compounds (F_(1,28)_ = 4.807, *p* < 0.05) (Figure 1D). However, comparison of CAT activity in the kidneys of the examined groups did not show significant differences between them (Figure 1D).

With regards to the GPx activity in the peripheral tissues, a two-way ANOVA performed in the liver revealed a significant treatment effect of BSO (F_(1,28)_ = 18.098, *p* < 0.001), a lack of GBR 12909 treatment effect (F_(1,28)_ = 0.004, NS) and no interaction between these two model compounds (F_(1,28)_ = 0.272, NS). Post hoc analysis showed that the GPx activity in this organ was significantly decreased in rats receiving both BSO alone and the BSO + GBR 12909 combination by 24.7 and 21.6% of the control value, respectively (Figure 1E). The same analysis carried out for the GPx activity in the kidneys demonstrated no treatment effects of both BSO (F_(1,28)_ = 0.152, NS) and GBR 12909 (F_(1,28)_ = 0.695, NS) and no interaction between the models compounds (F_(1,28)_ = 0.224, NS). GPx activity in the kidneys of these groups of rats remained at almost the same level as in the control (Figure 1F).

Regarding GR activity in the rat liver and kidneys (Figs. 1G, 1H), a two way ANOVA showed both significant treatment effects of BSO (for liver F_(1,28)_ = 28.855, *p* < 0.001; for kidneys F_(1,28)_ = 8.456, *p* < 0.01) and GBR 12909 (for liver F_(1,28)_ = 13.143, *p* < 0.002; for kidneys F_(1,28)_ = 15.309, *p* < 0.0001), as well as a significant interaction of these two compounds (for liver F_(1,28)_ = 25.888, *p* < 0.001; for kidneys F_(1,28)_ = 13.775, *p* < 0.001). In the liver, post hoc comparison of the studied groups showed that only the combined BSO + GBR 12909 administration dramatically increased GR activity by 110% of the control value (Figure 1G). In the kidneys, the same comparison showed that both BSO and GBR 12909 administered alone and in combination increased GR activity by 43.8%, 50.5% and 45.2% of the control value, respectively (Figure 1H).

### 3.2. The Effects of Chronic Treatment with BSO and GBR 12909 on the Enzymatic Activities of Superoxide Dismutase (SOD), Catalase (CAT), Glutathione Peroxidase (GPx) and Glutathione Disulfide Reductase (GR) in the Selected Brain Structures of 16-Day-Old Rats

As in the peripheral tissues, the changes in the enzymatic activity of SOD, CAT, GPx and GR were assessed in the chosen brain structures (prefrontal cortex, PFC; striatum, STR; and hippocampus, HIP), 4 h after the last chronic doses of BSO and GBR 12909, administered alone or in combination (Figure 2 and Figure 3).

As for SOD activity in the PFC and STR, a two-way ANOVA revealed significant treatment effects of BSO (for PFC F_(1,28)_ = 11.053, *p* < 0.01; for STR F_(1,28)_ = 6.540, *p* < 0.02) and GBR 12909 (for PFC F_(1,28)_ = 7.781, *p* < 0.01; for STR F_(1,28)_ = 15.118, *p* < 0.001), as well as an interaction between these model compounds (for PFC F_(1,28)_ = 8.300, *p* < 0.01; for STR F_(1,28)_ = 14.611, *p* < 0.001). Post hoc comparisons of the studied groups showed that in both the PFC and STR, SOD activity was significantly increased in groups of rats treated only with BSO (for PFC by 58.9%; for STR by 101.6% of control value) or GBR 12909 (for PFC by 53.8%; for STR by 122.8% of control value), as well as in the group receiving the combination of BSO + GBR 12909 (for PFC by 58%; for STR by 102.7% of control value) (Figure 2A,B). The same analysis performed for SOD activity in the HIP demonstrated no treatment effect of BSO (F_(1,28)_ = 3.289, NS), but a significant treatment effect of GBR 12909 (F_(1,28)_ = 13.016, *p* < 0.001) and no interaction between BSO and GBR 12909 (F_(1,28)_ = 1.503, NS) regarding this parameter. Post hoc comparisons showed that chronic treatment with GBR 12909 alone or jointly with BSO significantly decreased hippocampal SOD activity, but only vs. the BSO-treated group by 30.6% and 27.2%, respectively (Figure 2C).

As to the CAT activity in the studied brain structures (Figure 2D–F), a two-way ANOVA was significant only in the PFC. It revealed no treatments effect of both BSO (F_(1,28)_ = 4.187, *p* = 0.05) and GBR 12909 alone (F_(1,28)_ = 0.362, NS), but a significant interaction between BSO and GBR 12909 (F_(1,28)_ = 20.568, *p* < 0.001). In the PFC, post hoc comparisons showed that both BSO and GBR 12909 administered alone significantly increased CAT activity by 27.5% and 35.2% of the control value, respectively (Figure 2D). In the STR and HIP, a two-way ANOVA was insignificant, therefore no post-hoc comparisons were made between groups (Figure 2E,F).

Regarding the GPx activity in the studied brain structures, the most pronounced effects of BSO and GBR 12909 on this enzyme were observed in the STR. In this structure, a two way ANOVA revealed significant treatment effects of both BSO (F_(1,28)_ = 31.555, *p* < 0.00001) and GBR 12909 (F_(1,28)_ = 7.936, *p* < 0.01) and an interaction between these model compounds (F_(1,28)_ = 4.483, *p* < 0.05). Post hoc comparisons showed that BSO and GBR 12909 alone as well as the BSO + GBR 12909 combination reduced the activity of GPx by 26.6, 17 and 29% of control value, respectively (Figure 3B).

A two-way ANOVA for the GPx activity in the PFC was non-significant (Figure 3A), whereas in the HIP this analysis demonstrated no treatment effects of BSO (F_(1,28)_ = 3.056, NS) and GBR 12909 (F_(1,28)_ = 1.866, NS), but a significant interaction between these compounds (F_(1,28)_ = 6.305, *p* < 0.02). Post-hoc comparisons revealed that in the HIP of rats receiving the BSO + GBR 12909 combination, the GPx activity was markedly decreased only compared to the GBR-treated group, but it was comparable to that in the control and BSO-treated groups (Figure 3C).

With regards to the enzymatic activity of GR in the PFC and STR, a two-way ANOVA yielded non-significant results (Figure 3D,E), whereas in the HIP this analysis revealed treatment effects of BSO (F_(1,28)_ = 7.873, *p* < 0.01) and GBR 12909 (F_(1,28)_ = 22.115, *p* < 0.0001), and an interaction between these model compounds (F_(1,28)_ = 8.462, *p* < 0.01). Post hoc comparison showed that combined administration of BSO + GBR 12909 reduced the GR activity compared to the control, BSO- or GBR 12909-treated groups by 28.8, 28.8, 23.8%, respectively (Figure 3F).

### 3.3. The Effects of Chronic Administration of BSO and GBR 12909 on the Total Levels of ROS in the Liver, Kidney and the Selected Brain Structures

Comparison of the effects of model compounds on ROS levels in the liver showed that in the group treated with GBR 12909 alone, the total concentration of ROS was significantly reduced, but only vs. the group treated with BSO alone (Table 1). In turn in the kidney, post-hoc analysis showed that both BSO and GBR 12909, as well as their combined administration, significantly decreased the total ROS content compared to the control group by 44.8%, 57.7% and 52.4%, respectively (Table 1).

Regarding brain ROS concentrations, the post-hoc comparison showed no statistically significant differences between the groups treated with model compounds in any of the brain structures studied (Table 1).

### 3.4. The Effects of Chronic Administration of BSO and GBR 12909 on Lipid Peroxidation in the Liver, Kidney and the Selected Brain Structures

With regard to lipid peroxidation in the rat liver, a two-way ANOVA revealed a lack of treatment effect of BSO (F_(1,28)_ = 2.535, NS), but an overall treatment effect of GBR 12909 (F_(1,28)_ = 10.558, *p* < 0.01) and no interaction between these two compounds (F_(1,28)_ = 0.030, NS). Comparison of the examined groups showed that GBR 12909 in the liver decreased lipid peroxidation measured as the MDA level, but only vs. the BSO-treated group (Figure 4A).

The same analysis for lipid peroxidation in the kidneys demonstrated a significant treatment effect of BSO (F_(1,28)_ = 16.821, *p* < 0.001) but no treatment effect of GBR 12909 (F_(1,28)_ = 3.302, NS) and interaction between model compounds (F_(1,28)_ = 0.028, NS) (Figure 4B). Post-hoc analysis showed that administration of BSO alone significantly increased lipid peroxidation in the rat kidney, compared to the control group, while GBR 12909 given alone decreased it, but only vs. the BSO-treated group (Figure 4B).

In the studied brain structures, a two-way ANOVA showed, in the PFC, a significant treatment effects of both BSO (F_(1,28)_ = 5.342, *p* < 0.05) and GBR 12909 (F_(1,28)_ = 15.258, *p* < 0.001), but no interaction between these compounds (F_(1,28)_ = 0.09, NS), while in the STR and HIP, only a significant treatment effect of GBR 12909 (for STR F_(1,28)_ = 5.533, *p* < 0.05; for HIP F_(1,28)_ = 19.249, *p* < 0.0001), a lack of treatment effect of BSO (for STR F_(1,28)_ = 0.158, NS; for HIP F_(1,28)_ = 0.634, NS) and no interaction (for STR F_(1,28)_ = 0.329, NS; for HIP F_(1,28)_ = 0.006, NS). Comparisons of the examined groups within each structure showed that, in the PFC and HIP, the concentrations of MDA were significantly reduced both in rats treated with GBR 12909 alone and BSO + GBR 12909 combination when compared to the control (Figure 4C,E). In the STR there was no significant difference between the studied groups in the concentrations of MDA (Figure 4D).

### 3.5. The Effects of Chronic Treatment with BSO and GBR 12909 on the Levels of GSH, Cys and Met in the Striatum (STR)

As to GSH concentration in the STR, a two-way ANOVA revealed an overall treatment effect of BSO (F_(1,28)_ = 8.759, *p* < 0.01), but a lack of treatment effect of GBR 12909 (F_(1,28)_ = 0.688, NS), and no interaction between these two compounds (F_(1,28)_ = 0.016, NS). However, in the post-hoc test, there were no statistically significant differences in the content of GSH in the studied groups (Figure 5A).

A two-way ANOVA for Cys concentration in the STR was non-significant, the levels of this sulfur amino acid in the studied groups were similar (Figure 5B). The same analysis performed for Met content in this brain structure, revealed a significant treatment effect of GBR 12909 (F_(1,28)_ = 7.591, *p* < 0.01), but no treatment effect of BSO (F_(1,28)_ = 0.6, NS), and no interaction between these model compounds (F_(1,28)_ = 1.934, NS). Post-hoc analysis showed that the Met level in the GBR 12909-treated group was significantly higher than in the control group by 31.0% (Figure 5C).

### 3.6. The Effects of Chronic Treatment with BSO and GBR 12909 on the Boud Sulfane Sulfur Level in the Liver and Kidney

To check how chronic administration of model compounds affects the formation of bound sulfane sulfur, its level was measured only in the liver and kidneys, i.e., organs in which the highest decreases in GSH concentration were observed in our recently published study [34].

Regarding the bound sulfane sulfur in the liver, a two-way ANOVA revealed no treatment effect of BSO (F_(1,24)_ = 0.25, NS) but a significant effect of GBR 12909 (F_(1,24)_ = 9.264, *p* < 0.01), as well as an interaction between these compounds (F_(1,24)_ = 6.436, *p* < 0.02). Comparison of the studied groups showed that combined administration of BSO + GBR 12909 significantly decreased the level of bound sulfane sulfur vs. BSO-treated group (Figure 6A).

The same analysis for bound sulfane sulfur in the kidney demonstrated a lack of BSO treatment effect (F_(1,28)_ = 0.506, NS), a significant treatment effect of GBR 12909 (F_(1,28)_ = 4.822, *p* < 0.05) and a significant interaction between these compounds (F_(1,28)_ = 12.752, *p* < 0.001). Post-hoc analysis showed that both BSO and GBR 12909 significantly increased content of bound sulfane sulfur when compared to control group (Figure 6B).

## 4. Discussion

In our study, we performed a comprehensive analysis of the activity of the main antioxidant enzymes (SOD, CAT, GPx, GR), and the levels of ROS and MDA in the peripheral organs (liver, kidney) and selected brain structures (PFC, STR, HIP) of 16-day-old Sprague-Dawley rats chronically treated with model compounds (BSO and/or GBR 12909), and the data were related to the changes caused by these compounds in the concentrations of GSH and sulfur-containing amino acids, i.e., Met and Cys ([34], Table 2). In our study, due to technical reasons, the ROS levels were measured in frozen tissue homogenates, instead of freshly isolated tissue, which may cause some limitations. Therefore, in order to confirm the effects of oxidants in the examined tissues, the levels of MDA (for which the freezing of tissue is not relevant) were determined as a measure of lipid peroxidation.

To facilitate tracing these complicated relationships between the studied parameters and taking into account the specificity of metabolism of sulfur-containing compounds in different tissues [34], the discussion was divided into subsections.

### 4.1. Basic Activity of Antioxidant Enzymes and Production of ROS in the Studied Tissues

In general, the examined tissues differ in the level of basic activity of antioxidant antioxidant enzymes. Compared to peripheral organs, the brain is characterized by a relatively low activity of these enzymes [54,55,56,57,58]. Furthermore, both in peripheral tissues and in the brain, the activities of these enzymes were significantly lower during early postnatal life than in adulthood [54]. In our study, performed on 16-day-old rats, the activity of SOD, CAT and GPx in the liver of the control group was many times higher than in the studied brain structures (see Figure 1). Only the basic activity of GR in the liver was comparable to that in the brain. In turn, in the kidneys of the control group, CAT, GPx and GR activities were significantly higher than in the brain, and only SOD activity remained at a similar level in both organs.

Despite a relatively low basic activity of antioxidant enzymes, brain cells are characterized by very high oxygen consumption. In fact, the brain metabolizes almost 20% of the total pool of body oxygen, although it constitutes hardly 2% of the body weight. Due to such a high oxygen demand, brain mitochondria generate large amounts of ROS, such as O_2_^•^^−^ and H_2_O_2_ as a consequence of aerobic metabolism and ATP synthesis [35,59,60,61]. The activity of enzymes belonging to the NADPH oxidase (NOX) family is another major source of ROS in the central nervous system [35,62]. These enzymes contribute to maintaining the physiological ROS level according to cellular demands [62,63]. The primary product of NOX activity O_2_^•^^−^ is immediately converted into H_2_O_2_ by the SOD enzyme that is physically associated with NOX. In cultured hippocampal neurons, the inhibition of NOX enzymes has been shown to reduce intracellular H_2_O_2_ levels by almost 45% [63]. It indicates that NOX enzymes are the main component responsible for the intracellular H_2_O_2_ levels and, together with mitochondria, for maintaining the oxidative power of the cell [35,62,64]. Other sources of ROS in the brain include xanthine oxidases, lipoxygenases, nitric oxide synthases and catecholamines [59,65,66].

In the developing brain, ROS act as important regulators of various physiological processes, such as cell proliferation and differentiation, via controlling different signal pathways, a process defined as redox signaling [42,63,67]. In light of the above data, it seems reasonable to assume that the significantly lower activity of antioxidant enzymes in the brain than in peripheral tissues may be adaptive in nature, due to a high demand of brain cells for ROS to be used for cell signaling.

### 4.2. Potential Cause-Effect Relationships between Concentrations of GSH, Met, ROS and Lipid Peroxidation Products and the Antioxidant Enzymes Activities in the Liver, Kidney and Brain of Rats Treated with BSO and GBR 12909, Alone and in Combination

#### 4.2.1. Liver

Referring to the ROS production in our study, administration of BSO and GBR 12909, alone or in combination, did not cause significant increases in the total ROS levels in any of the studied tissues. However, a significant decrease in the total ROS content was observed in the liver of rats receiving GBR 12909 alone, compared to rats treated with BSO alone. Along with the decrease in the total ROS content in this group, a significant decrease in the MDA concentration, which is a measure of lipid peroxidation, was also observed. Parallel to the decrease in ROS and MDA concentrations, a significant increase in the level of Met was found in the GBR 12909-treated group (Table 2). It is worth noting that in this group, the increase in Met content was much higher than in the group treated with BSO alone, but comparable to that in the group receiving BSO + GBR 12909 in combination (Table 2). In addition, there was a small but significant increase in the GSH level in the GBR 12909-treated group. The above data clearly suggest that in the liver of GBR 12909-treated group, the total ROS level and subsequent lipid peroxidation may be regulated by both Met and GSH [68,69,70,71]. However, in the BSO− or BSO + GBR 12909 treated groups, in which there was a drastic reduction in GSH levels with simultaneous high increases in Met concentrations, the levels of ROS and MDA did not differ significantly, compared to the control group. The latter effect seems to indicate that under conditions of a strong inhibition of GSH synthesis, the total ROS level and subsequent lipid peroxidation may be controlled mainly by Met.

Met is one of the most easily oxidized amino acids by ROS, being converted to methionine sulfoxide (MetO) [72]. Free Met molecules, as well as Met residues in proteins, are efficient scavengers of almost all oxidizing molecules, such as H_2_O_2_, hydroxyl radical, peroxynitrite and hypochlorous acid [73,74,75]. A non-polar S-methyl thioeter side chain of Met is almost always situated in the interior hydrophobic core of globular proteins. However, as a component of membrane-spanning proteins, the Met side chain can be located on the protein surface, acting as an endogenous antioxidant [73]. During H_2_O_2_ removal, Met is transformed into MetO, which can be reduced back to Met, in a reaction catalyzed by Met sulfoxide reductases (Msr) in the presence of nicotinamide adenine dinucleotide phosphate (NADPH) [71,73,74,76]. Since two MetO stereoisomers are formed during the Met oxidation, called MetO-S and MetO-R, they are reduced back to Met by two Msr enzymes, i.e., Msr-A and Msr-B, respectively [77,78,79]. Cyclic oxidation and reduction of MetO both scavenges H_2_O_2_ and drives an NADPH oxidation reaction. Furthermore, it has been recently demonstrated that Met activates the Msr-A [70]. Thus, Msr enzymes represent repair mechanisms that can reduce the oxidized Met via enzymatic reaction in this way decreasing the oxidative stress [76,80,81]. Therefore, further research is needed to check whether the increase in Met levels in the rat liver caused by the model compounds will increase Msr-A activity, and consequently will intensify the MetO reduction reaction.

Analyzing the role of antioxidant enzymes in the regulation of ROS concentrations, especially H_2_O_2_, it is important to note, a significant reduction in the GPx activity was observed in the liver of rats treated with BSO alone or in combination with GBR 12909. The decreases in GPx activity in these groups could be explained by GSH deficiency, which is the substrate for H_2_O_2_ removal by this enzyme. However, an explanation of the decrease in CAT activity, which, regardless of the presence of GSH, also removes H_2_O_2_, mainly in peroxisomes [82], in the group of rats treated with GBR 12909 alone is not so simple. Therefore, it seems likely that the observed changes in the GPx activity in the groups treated with BSO/BSO + GBR 12909, and in the CAT activity in the group receiving GBR 12909 alone, may reflect an adaptive mechanism aimed at maintaining adequate H_2_O_2_ levels for cell signaling. Regarding SOD, an enzyme that scavenging O_2_^•−^ produces H_2_O_2_, its activity in the liver of the studied groups of rats remained at a similar level. Interestingly, the increased SOD activity and reduced ROS content were observed in the liver of rats receiving high doses of Met [80].

As for the activity of GR, the enzyme that transfers electrons from NADPH to glutathione disulfide (GSSG) thereby regenerating the GSH pool, in our study, its activity in the rat liver increased significantly only in the group treated with the BSO + GBR 12909 combination, in which the decline in GSH and concomitant increase in Met concentrations were the largest. It means that, particularly in this group of rats, the demand for GSH and NADPH increased. NADPH is needed for the enzymatic activity of both GR and Msr [83], so the availability of NADPH will be a factor limiting the activity of these enzymes at conditions of the combined treatment with BSO + GBR 12909. Therefore, further studies are needed to check whether the activity of the pentose cycle in which NADPH is formed, increases in the liver of this group of rats.

#### 4.2.2. Kidney

In contrast to the liver, in the kidneys of rats treated with BSO, alone or in combination with GBR 12909, the total ROS levels were significantly reduced. In these groups, the decreases in GSH and concomitant increases in Met content were weaker than in the liver (Table 2). In addition, in the kidneys of rats treated with GBR 12909 alone—in which GSH level was unchanged, while the Met showed only an increasing tendency (Table 2)—the total ROS content was also reduced to a similar extent as in the groups treated with BSO. It is difficult to explain such strong decreases in the total ROS levels in the studied groups only on the basis of the increased Met contents. These data indicate the existence of some additional mechanisms modulating the total content of ROS in this organ. It is worth noting that the kidney is a special organ characterized by extremely high activity of γ-glutamyl transpeptidase (γ-GT), the only enzyme located on the outer surface of the cell membrane capable of breaking the unusual γ-peptide bond within the GSH molecule [84,85], which is mainly responsible for maintaining a particularly high renal Cys concentration [52,86]. During anaerobic metabolism of Cys, sulfane sulfur-containing compounds are formed that have strong antioxidant properties [87,88,89]. Due to the highest level of Cys in the kidneys, the concentration of sulfane sulfur-containing compounds in this organ is also much higher than in other tissues [90]. Although it has previously been shown that, in the kidneys of rats treated with BSO alone or in combination with GBR 12909, Cys levels were significantly reduced compared to the control group, they were still markedly higher than in the liver [34]. This means that the formation of sulfane sulfur compounds should still be high in these groups. This assumption is consistent with our additional data, showing that in the kidneys of rats treated with BSO or GBR 12909 alone the levels of the bound sulfane sulfur were significantly higher than in controls. The bound sulfane sulfur mainly includes hydropersulfides (RSSH), which are much more powerful radical scavengers than thiols [88,91]. Hence, the decrease in ROS levels in the kidneys of the studied groups of rats could be explained by increases in both Met and bound sulfane sulfur levels.

On the other hand, our results also revealed that in the kidneys of rats treated with BSO alone, despite an increase in the level of bound sulfane sulfur and a decline in the total ROS level, an increase in the lipid peroxidation measured as the MDA level was observed. It is well known that reactive sulfur species (RSS), including RSSH, can be reduced with the release hydrogen sulfide (H_2_S), which can then be oxidized to sulfite. It has been previously shown that sulfite radicals can initiate lipid peroxidation [92]. The sulfite can also be formed directly from cysteine sulfinic acid, which is the product of Cys oxidation by cysteine dioxygenase (CDO). In the BSO-treated group, in our study, the determined level of GSH was 2-fold lower than in the control group, which could suggest accumulation of Cys not used for GSH synthesis. However, the measured Cys level in the BSO-treated group was significantly reduced, when compared to the control group. This implies that Cys was effectively metabolized by CDO, which, in consequence, could lead to the production of sulfite radicals, and ultimately to lipid peroxidation. A similar effect was described in our earlier study [93]. In the latter study, repeated treatment with cocaine did not change the total ROS level in the kidney, but caused a significant increase in lipid peroxidation, which was accompanied by a simultaneous increase in bound sulfane sulfur level [93]. Both results suggest that RSS, whose concentration is particularly high in the kidney, under certain conditions, may be responsible for increased lipid peroxidation.

Regarding the activity of antioxidant enzymes in the kidneys, significant increases in SOD activity were observed in the groups of rats treated with BSO or GBR 12909 alone. An increase in the SOD activity could be interpreted as a defense against the oxidative stress caused by excessive production of O_2_^•−^. However, the lack of concomitant increases in the activity of GPx and CAT that remove H_2_O_2_, may also suggest an increased demand for H_2_O_2_, presumably for signaling purposes. This supposition seems particularly justified, because the observed decrease in the total ROS level in all studied groups also means a decrease in the H_2_O_2_ pool.

However, the most intriguing effect of BSO and GBR 12909, administered alone or jointly, was related to an increase in GR activity in the kidneys of all studied groups of rats. GR is an NADPH-dependent enzyme that, in addition to GSSG, also reduces other disulfides, such as l-cystine (Cys)_2_, di-γ-glutamylcystine and DL-homocystine [94]. Di-γ-glutamylcystine is formed in large amounts in the kidney during extracellular degradation of GSH by γ-GT when the γ-glutamyl moiety released by this enzyme is transferred to the amino acid acceptor, i.e., (Cys)_2_. Then di-γ-glutamylcystine, after being taken up into cells, is reduced to γ-glutamylcysteine (γ-GluCys), a dipeptide, which is normally formed during the first step of GSH synthesis, that is, the ATP-dependent reaction catalyzed by GCL. Just this reaction was blocked in our study by BSO. The second step of GSH synthesis, i.e., the ATP-dependent reaction catalyzed by glutathione synthetase during which γ-GluCys is coupled with glycine thus forming GSH [95], is not inhibited by BSO. So, in the kidney, the alternative pathway of γ-GluCys formation allows for bypassing the first reaction catalyzed by GCL and for continuation of GSH synthesis. Due to the specificity of GSH synthesis in the kidney, the observed decreases in the GSH content in rats treated with BSO, alone or in combination with GBR 12909, were smaller than in the liver [34]. Thus, a particularly significant increase in GR activity in the kidneys of rats treated with BSO and GBR 12909, alone or in combination, appears to result from the intensification of reduction reaction of some disulfides, e.g., di-γ-glutamylcystine to maintain GSH synthesis.

#### 4.2.3. Brain

Regarding the concentration of ROS in the examined brain structures, only in the HIP of rats treated with GBR 12909 alone, their level was significantly reduced compared to the control group. In response to the decrease in the total ROS level, a significant decline in the concentration of MDA, i.e., a marker of lipid peroxidation, was observed in this structure. In parallel, a small, but significant increase in GSH (by 8.5% of the control) and a relatively high increase in Met (by 62.7% of the control) concentration were found in this group ([34], Table 2). The latter effects suggest that in the HIP of GBR 12909-treated group, both the enhanced GSH level, and a particularly high content of free Met may be responsible for scavenging ROS and finally decreasing the level of lipid peroxidation below the control value. However, the PFC of the GBR-treated group, in which the ROS and GSH concentrations remained at the control levels, and Met showed barely an upward trend, the Cys content was significantly increased, while the MDA level was markedly reduced compared to the control group. The latter effect implicates that the elevated level of Cys may play a significant role in the decline of the lipid peroxidation level in the PFC of this group of rats. In our study, only the total level of free, non-protein Cys (reduced + oxidized forms) was determined, however, in cells, Cys exists mainly in the reduced form, in contrast to the extracellular space, where it occurs as a disulfide [96,97]. It is also worth adding that L-Cys is a major substrate to produce about 70% of endogenous H_2_S that is a potent reducing agent [98,99]. Hence, the observed increase in the total level of Cys may indicate an increase in reducing power in cells, which may result in a decrease in the lipid peroxidation level.

Furthermore, significant increases in the total Cys levels were also observed in the PFC and HIP of rats receiving both BSO alone and the BSO + GBR 12909 combination. However, only in the latter group, were significant decreases in the levels of lipid peroxidation observed, both in the PFC and HIP. Interestingly, although in the PFC a decrease in the lipid peroxidation level occurred only in the BSO + GBR 12909-treated group, but not in the group receiving BSO alone, increases in the Cys content were comparable in both groups. Therefore, it seems likely that in the PFC and HIP of the BSO + GBR 12909-treated group, besides Cys, GBR 12909 itself may also affect lipid peroxidation. This assumption is consistent with the study by Camarero et al. [100], who showed that GBR 12909 was able not only to inhibit dopamine reuptake, but also to reduce MDA formation, although this compound had no significant intrinsic radical trapping activity. The above-presented comparative analysis of the potential factors affecting lipid peroxidation in the selected brain structures clearly indicates that GBR 12909, used as a model compound, can modify not only dopaminergic transmission, but also lipid peroxidation. Transient changes in the expression of both these parameters during early postnatal development may have long-term consequences that become apparent in adulthood.

In contrast to the reduced lipid peroxidation level in the PFC and HIP of BSO + GBR-treated Sprague-Dawley pups in our study, other researchers found increased lipid peroxidation in diencephalon and pons/medula in ODS pups, but no changes in lipid peroxidation in other brain structures in both ODS and non-mutant OFA pups receiving either BSO alone or the BSO + GBR 12909 combination [21,101]. It is difficult to explain the differences between these results, especially because, in the cited studies, the levels of sulfur-containing amino acids (Met, Cys) were not determined. In addition, there are more such differences in the tested parameters between the rat strains used. Although the doses and the administration regimen of BSO and GBR 12909 were the same, our study found a small but statistically significant decrease in GSH content only in the PFC, while in other studies performed on ODS, non-mutant OFA and Wistar rats, the decreases in GSH levels were deeper (by 20% to 50% of control value), and occurred in various brain structures [22,101]. The reason for the discrepancy in GSH levels in the above-discussed studies is unclear, but it is supposed that it may be due to some specific characteristics of the studied rat strains (e.g., genetic background, blood brain barrier permeability, rate of metabolism).

As for the activity of antioxidant enzymes in the selected brain structures, the most characteristic feature is the increase in SOD activity in the PFC and STR of BSO and/or GBR12909-treated groups. In the PFC, the increased SOD activity was not accompanied by any changes in GPx and GR activities, and only the enhanced CAT activity was observed in the groups treated with BSO or GBR 12909 alone. The increased CAT activity in the latter groups suggests that H_2_O_2_ production by SOD was, at least to some extent, controlled by CAT. In contrast to our results, in the PFC of ODS mutant rats no changes in SOD and CAT activities were found in BSO and/or GBR12909-treated groups [101]. As to SOD activity in the STR in our study, the observed increase in its activity with a simultaneous decrease in GPx activity and no change in CAT activity seems to indicate that in this structure, H_2_O_2_ deficiency in the total ROS pool may have occurred. Finally, in the HIP of the BSO + GBR 12909-treated group, there was no significant changes in the SOD, GPx and CAT activities when compared to control. Only the GR activity was significantly reduced, compared to the control, BSO- and BSO + GBR 12909-treated groups. The decrease in GR activity in this group of rats could be interpreted as an adaptive change, resulting from an excess of reducing agents.

## 5. Conclusions

A comprehensive analysis of the levels of ROS, MDA and the activity of the major antioxidant enzymes (SOD, CAT, GPx, GR) performed in groups of rats treated with BSO alone or the BSO + GBR 12909 combination, with respect to changes in both the contents of GSH and sulfur amino acids (Met and Cys), showed that only in the BSO + GBR 12909-treated group a significant decrease in lipid peroxidation occurred in the PFC and HIP (Figure 7). Lipids are important components of cell membranes, hence decreases in the levels of their oxidation in the PFC and HIP, i.e., the brain structures playing an important role in the pathogenesis of schizophrenia during early postnatal development may mean changes in cellular redox signaling, which may contribute to the appearance of schizophrenia-like symptoms in adulthood. Consistently with this supposition, in our previous study [34], only in the group of rats receiving the BSO + GBR 12909 combination during early postnatal life, did behavioral changes resembling positive symptoms in schizophrenia patients occur in adulthood. However, both in rats treated with BSO alone or BSO + GBR 12909 during development, the schizophrenia-like deficits in social behaviors and cognitive functions were observed in adulthood. Therefore, further studies are needed to clarify whether changes in the lipid peroxidation in the PFC and HIP of rats treated with BSO + GBR 12909 combination affect intracellular signaling pathways.

## Figures and Tables

**Figure 1 antioxidants-09-00538-f001:**
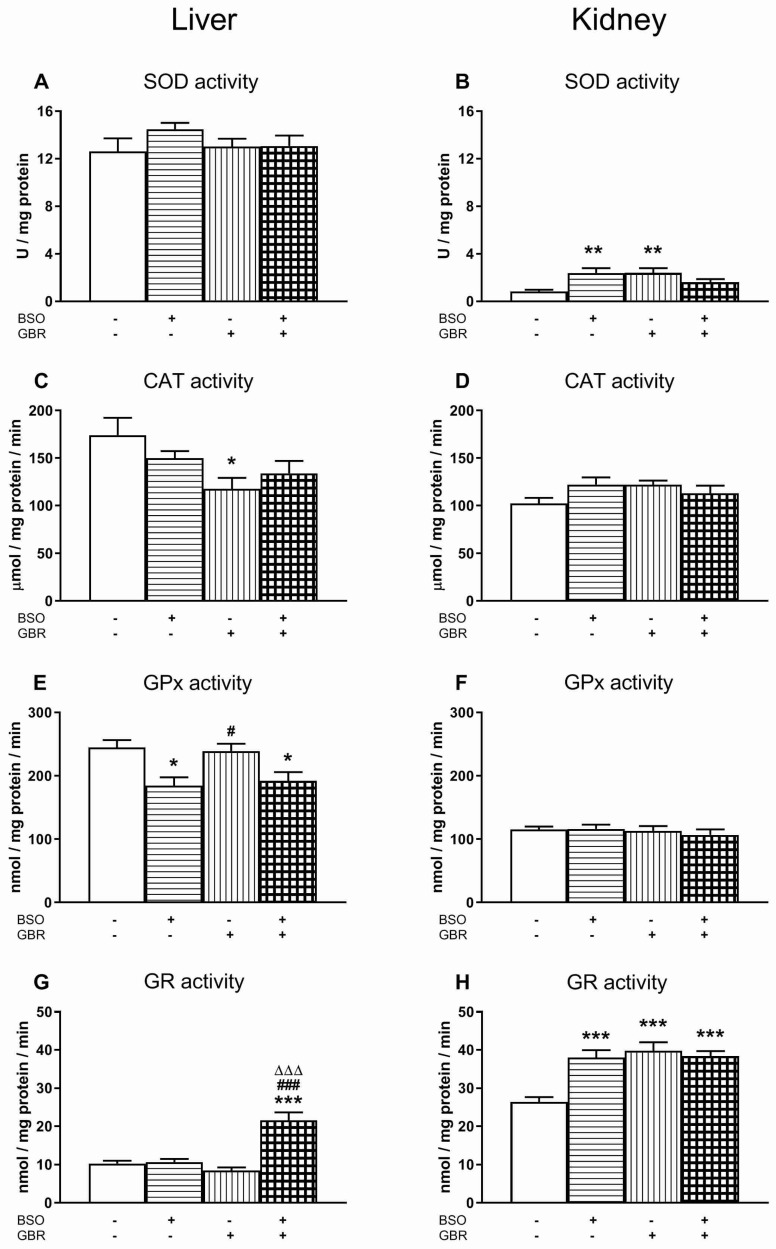
The effect of chronic administration of l-Buthionine-(S,R)-sulfoximine (BSO) and/or 1-[2-[Bis(4-fluorophenyl)methoxy]ethyl]-4-(3-phenylpropyl)piperazine dihydrochloride (GBR 12909) on the enzymatic activities of superoxide dismutase (SOD; 1**A** and 1**B**), catalase (CAT; 1**C** and 1**D**), glutathione peroxidase (GPx; 1**E** and 1**F**) and glutathione-disulfide reductase (GR; 1**G** and 1**H**) in the liver and kidney of rats. SOD activity is expressed in U/mg of protein (the amount of enzyme needed to produce 50% dismutation of O_2_^•−^ radical), the activity of CAT is expressed in μmol/mg of protein/min (μmol of H_2_O_2_ degraded by the enzyme during 1 min per mg of protein). GPx and GR activities are expressed in nmol/mg of protein/min (the amount of enzyme that causes the oxidation of 1 nmol of nicotinamide adenine dinucleotide phosphate (NADPH) to NADP^+^ during 1 min per mg of protein). The oxidation of NADPH to NADP^+^ is directly proportional to GPx or GR activities in the samples. The bars representing activities of particular enzymes in the studied groups show the mean ± SEM, *n* = 8. Symbols indicate significance of differences according to the Bonferroni post-hoc test, ^*^
*p* < 0.05, ^**^
*p* < 0.01, ^***^
*p* < 0.001 vs. control-; ^#^
*p* < 0.05, ^###^
*p* < 0.001 vs. BSO- and ^∆∆∆^
*p* < 0.001 vs. GBR 12909-treated groups.

**Figure 2 antioxidants-09-00538-f002:**
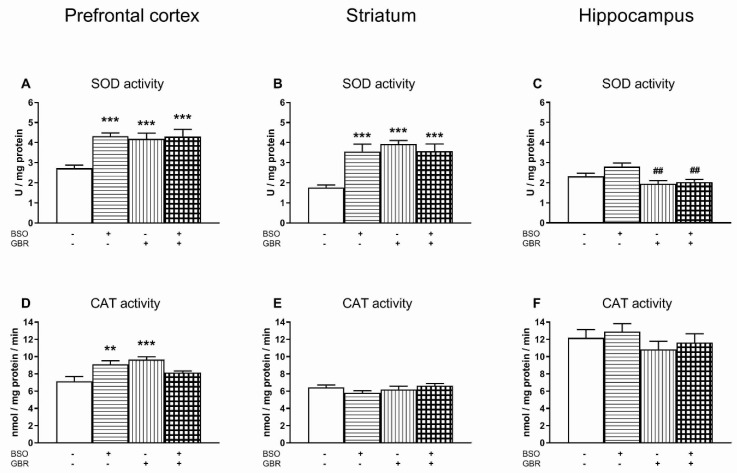
The effect of chronic administration of BSO and/or GBR 12909 on the enzymatic activities of superoxide dismutase (SOD; **2A–2C**) and catalase (CAT; **2D–2F**) in the prefrontal cortex, striatum and hippocampus of 16-day-old rats. Activity of SOD is expressed in U/mg of protein (the amount of enzyme needed to produce 50% dismutation of O_2_^•−^ radical), the activity of CAT is expressed in nmol/mg of protein/min (the amount of enzyme that causes the formation of 1 nmol of formaldehyde during 1 min per mg of protein). The bars representing SOD or CAT activity in particular groups show the mean ± SEM, *n* = 8. Symbols indicate significance of differences according to the Bonferroni post-hoc test, ^**^
*p* < 0.01, ^***^
*p* < 0.001 vs. control- and ^##^
*p* < 0.01 vs. BSO-treated groups.

**Figure 3 antioxidants-09-00538-f003:**
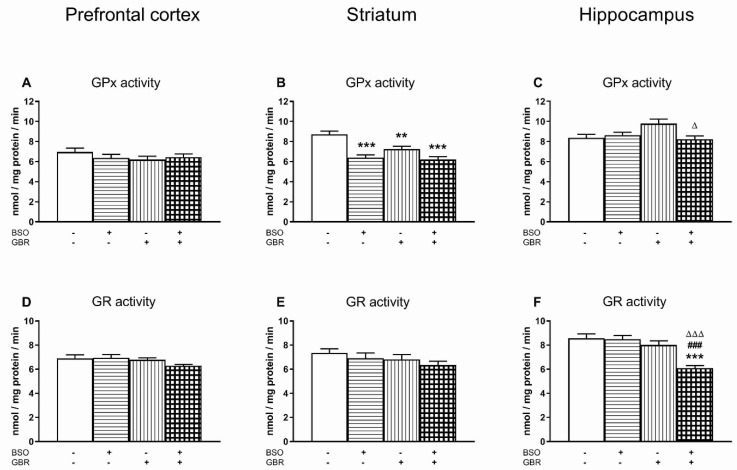
The effect of chronic administration of BSO and/or GBR 12909 on the enzymatic activities of glutathione peroxidase (GPx; **3A–3C**) and glutathione-disulfide reductase (GR; **3D–3F**). The enzymatic activities of GPx and GR are expressed in nmol/mg protein/min (nmol of NADPH oxidized to NADP^+^ by the enzyme during 1 min per mg of protein). The bars representing GPx or GR activity in particular groups show the mean ± SEM, *n* = 8. Symbols indicate significance of differences according to the Bonferroni post-hoc test, ^**^
*p* < 0.01, ^***^
*p* < 0.001 vs. control-; ^###^
*p* < 0.001 vs. BSO- and ^∆^
*p* < 0.05, ^∆∆∆^
*p* < 0.001 vs. GBR 12909-treated groups.

**Figure 4 antioxidants-09-00538-f004:**
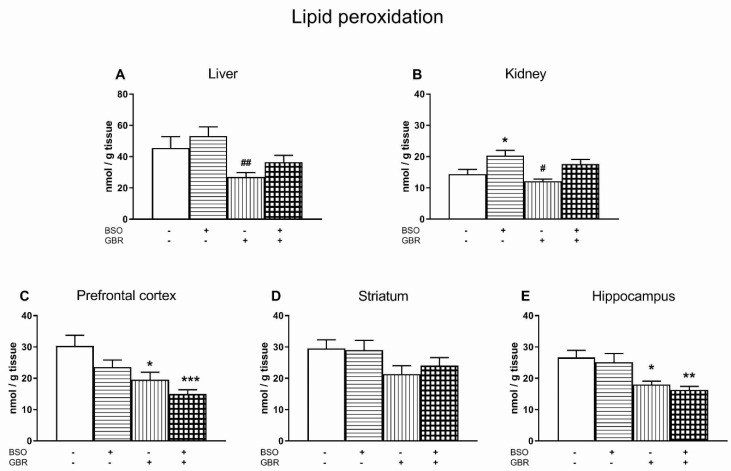
The effects of chronic administration of BSO and/or GBR 12909 on lipid peroxidation expressed as the level of malondialdehyde (MDA) in the liver (**A**), kidney (**B**), prefrontal cortex (**C**), striatum (**D**) and hippocampus (E) of pups. Data expressed in nmol/g of tissue are presented as the mean ± SEM, *n* = 8 for each group. Symbols indicate significance of differences, according to the Bonferroni post-hoc test, ^*^
*p* < 0.05, ^**^
*p* < 0.01, ^***^
*p* < 0.001 vs. control; ^#^
*p* < 0.05, ^##^
*p*< 0.05 vs. the BSO-treated group.

**Figure 5 antioxidants-09-00538-f005:**
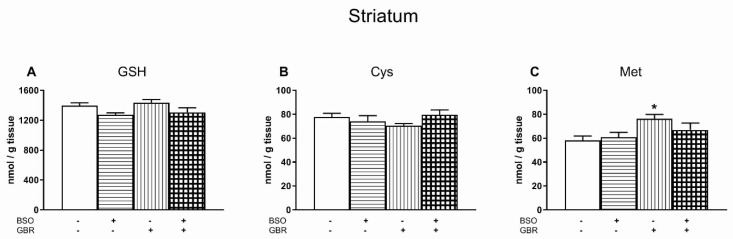
The effects of chronic administration of BSO and/or GBR 12909 on the levels of GSH (**A**), Cys (**B**) and Met (**C**) in the striatum (STR) of pups. Data expressed in nmol/g of tissue are presented as the mean ± SEM, *n* = 8 for each group. Symbols indicate significance of differences according to the Bonferroni post-hoc test, ^*^
*p* < 0.05 vs. the control group.

**Figure 6 antioxidants-09-00538-f006:**
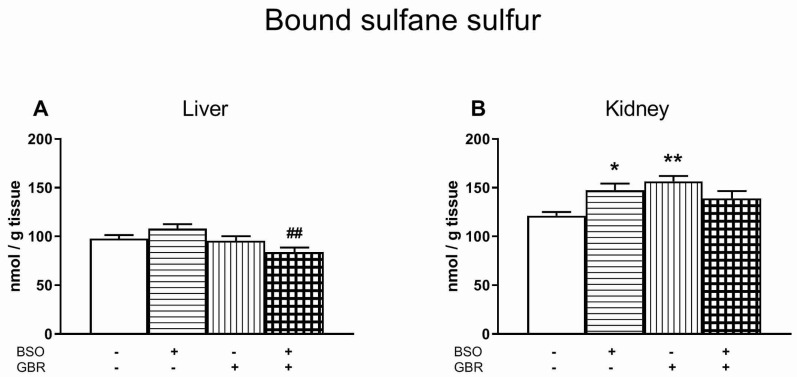
The effects of chronic administration of BSO and/or GBR 12909 on the level of bound sulfane sulfur in the liver (**A**) and kidney (**B**) of 16-day-old rats. Data expressed in nmol/g of tissue are presented as the mean ± SEM, *n* = 7 for liver, *n* = 8 for kidney. Symbols indicate significance of differences according to the Bonferroni post-hoc test, ^*^
*p* < 0.05, ^**^
*p* < 0.01 vs. control; ^##^
*p* < 0.01 vs. the BSO-treated group.

**Figure 7 antioxidants-09-00538-f007:**
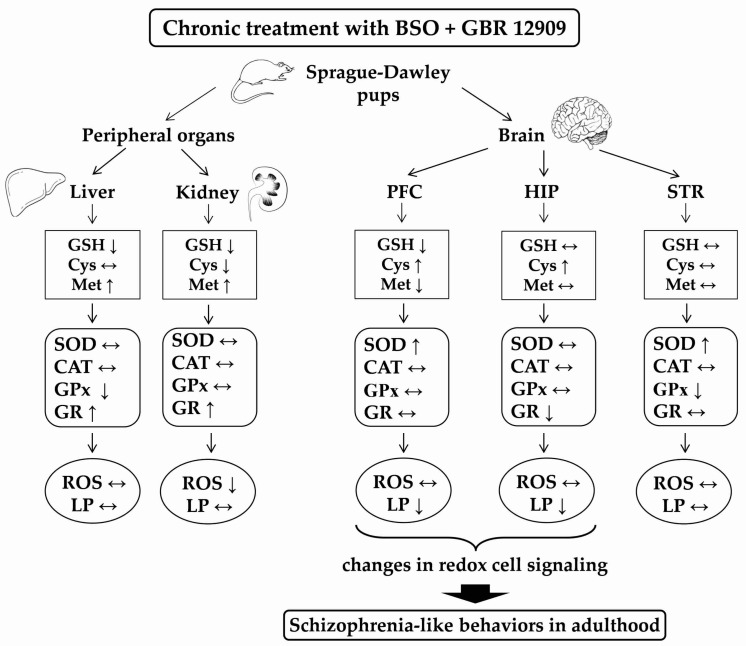
Graphic illustration of the observed changes in GSH, Cys and Met concentrations, antioxidant enzyme activities (SOD, CAT, GPx, GR), the total levels of ROS and lipid peroxidation (LP) in peripheral tissues (liver, kidneys) and selected brain structures (PFC, HIP, STR) of Sprague-Dawley pups treated at postnatal days 5-16 with BSO + GBR 12909 combination. Arrows show: **↑** – increase; **↓** – decrease; and **↔** – no changes in the studied parameters.

**Table 1 antioxidants-09-00538-t001:** The effects of chronic treatment with l-Buthionine-(S,R)-sulfoximine (BSO) and/or 1-[2-[Bis(4-fluorophenyl)methoxy]ethyl]-4-(3-phenylpropyl)piperazine dihydrochloride (GBR 12909), alone and in combination, on the total reactive oxygen species (ROS) levels in the peripheral organs and the selected brain structures of 16-day-old Sprague-Dawley rats.

Tissue	Experimental Groups	ROS (nmol/g Tissue)	ANOVA Results
Peripheral Organs			
Liver	control BSO GBR BSO + GBR	48.6 ± 7.8 67.9 ± 9.0 38.7 ± 3.7 ^#^ 53.2 ± 3.6	Effect of BSO F(1,24) = 6.789, *p* < 0.05 Effect of GBR F(1,24) = 3.556, *p* = 0.07 Interaction F(1,24) = 0.133, NS
Kidney	control BSO GBR BSO + GBR	89.8 ± 9.7 49.7 ± 3.1 *** 38.0 ± 5.5 *** 42.8 ± 6.1 ***	Effect of BSO F(1,28) = 7.320, *p* < 0.05 Effect of GBR F(1,28) = 19.979, *p* < 0.001 Interaction F(1,28) = 11.776, *p* < 0.01
Brain Structures			
Prefrontal Cortex	control BSO GBR BSO + GBR	38.7 ± 6.3 36.3 ± 6.6 39.6 ± 7.2 38.4 ± 7.6	Effect of BSO F(1,24) = 0.073, NS Effect of GBR F(1,24) = 0.045, NS Interaction F(1,24) = 0.008, NS
Hippocampus	control BSO GBR BSO + GBR	24.8 ± 1.1 23.2 ± 1.6 19.9 ± 1.1 23.8 ± 1.0	Effect of BSO F(1,24) = 0.946, NS Effect of GBR F(1,24) = 3.246, *p* = 0.08 Interaction F(1,24) = 4.957, *p* < 0.05
Striatum	control BSO GBR BSO + GBR	28.2 ± 5.4 19.7 ± 1.9 18.4 ± 2.6 20.8 ± 2.9	Effect of BSO F(1,24) = 0.768, NS Effect of GBR F(1,24) = 1.583, NS Interaction F(1,24) = 2.517, NS

The obtained data expressed in nmol/g of tissue are presented as the mean ± SEM, *n* = 7 for the liver and all studied brain structures, *n* = 8 for kidneys. Symbols indicate significance of differences according to the Bonferroni post-hoc test, ^***^
*p* < 0.001 vs. control; ^#^
*p* < 0.05 vs. BSO-treated group.

**Table 2 antioxidants-09-00538-t002:** The effects of chronic treatment with BSO and/or GBR 12909 on the levels of glutathione (GSH), cysteine (Cys) and methionine (Met) in the peripheral organs and selected brain structures of 16-day-old Sprague-Dawley rats according to Górny et al. [34].

Tissue	Experimental Groups	GSH	Cys	Met
Peripheral Organs				
Liver	control BSO GBR BSO + GBR	100% ↓ by 66.7% ↑ by 18.7% ↓ by 76.4%	100% ↔ ↓ tendency ↔	100% ↑ by 114% ↑ by 315% ↑ by 381.5%
Kidney	control BSO GBR BSO + GBR	100% ↓ by 49% ↔ ↓ by 30%	100% ↓ by 43% ↔ ↓ by 41%	100% ↑ by 32% ↑ tendency ↑ by 27%
Brain Structures				
Prefrontal cortex	control BSO GBR BSO + GBR	100% ↓ by 7% ↔ ↓ by 7%	100% ↑ by 47.5% ↑ by 28.5% ↑ by 42.2%	100% ↓ by 18.5% ↑ tendency ↓ by 19.7%
Hippocampus	control BSO GBR BSO + GBR	100% ↔ ↑ by 8.5% ↔	100% ↑ by 34.9% ↑ tendency ↑ by 30.9%	100% ↑ by 37.5% ↑ by 62.7% ↔

Concentrations of GSH, Cys and Met are expressed as a percentage of the appropriate control value. Arrows indicate: ↑ – increase; ↓ – decrease; ↔ no changes.

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
