# Peer review of "Alterations in the Antioxidant Enzyme Activities in the Neurodevelopmental Rat Model of Schizophrenia Induced by Glutathione Deficiency during Early Postnatal Life"

_antioxidants, 2020, doi:10.3390/antiox9060538_

Round 1
Reviewer 1 Report
The paper has been significantly improved from the last submission.
The results are clear and well discussed. The graphical illustration in Figure 7 facilitates the reading and the overall discussion.
Minor: Table 2 needs reformatting (arrows are overlaying the text)
Author Response
Reply to Reviewer 1
Reviewer's comment
The paper has been significantly improved from the last submission.
The results are clear and well discussed. The graphical illustration in Figure 7 facilitates the reading and the overall discussion.
Minor: Table 2 needs reformatting (arrows are overlaying the text).
Reply to Reviewer's comment
Thank you very much for the positive evaluation of the revised version of the manuscript.
Table 2 has been corrected in the submitted version of the work (proof). Table was prepared according to the Instruction for authors using the Table option of Microsoft Word to create tables. To avoid moving the arrows in the attached table, we additionally send this table in high resolution PDF format.

Reviewer 2 Report
The article has been greatly improved in this revised version which takes into account most comments and suggestions (including additionnal data regarding lipid peroxidation and bound sulfane sulfur). The last summary figure is also very useful. At this point, I would recommend publication of this study.
However, I still feel unease about the ROS measurements in frozen tissue homogenates and its interpretation . I believe that clearly acknowledging and briefly discussing this limitation (along the same line of what the authors responded in their letter to the editor) would be scientifically appropriate. It would also further set the stage for the additional measurements of lipid peroxidation.
Here are still few minor corrections:
line 91: ".. administered chronically with BSO..." (with is missing)
line 373: The Cat activity in striatum and hippocampus is not presented in the text (only in figure 3). Is it on purpose ?
Lines 391-393: description of the postdoc analysis does not fit with the significance indicated in Fig.3C
Table1: the lines of the column "experimental groups" are not at the same level that the corresponding lines in the other columns.
Table 2: the arrows are misplaced in the table
In figure 7: why not also add the bound sulfane sulfur for liver and kidney ?
Author Response
Reply to Reviewer 2
Reviewer's Comment and Suggestions for Authors
The article has been greatly improved in this revised version which takes into account most comments and suggestions (including additionnal data regarding lipid peroxidation and bound sulfane sulfur). The last summary figure is also very useful. At this point, I would recommend publication of this study.
Reviewer's general comments
However, I still feel unease about the ROS measurements in frozen tissue homogenates and its interpretation. I believe that clearly acknowledging and briefly discussing this limitation (along the same line of what the authors responded in their letter to the editor) would be scientifically appropriate. It would also further set the stage for the additional measurements of lipid peroxidation.
Reviewer's minor comments
Here are still few minor corrections:
line 91: ".. administered chronically with BSO..." (with is missing)
line 373: The Cat activity in striatum and hippocampus is not presented in the text (only in figure 3). Is it on purpose ?
Lines 391-393: description of the postdoc analysis does not fit with the significance indicated in Fig.3C
Table1: the lines of the column "experimental groups" are not at the same level that the corresponding lines in the other columns.
Table 2: the arrows are misplaced in the table
In figure 7: why not also add the bound sulfane sulfur for liver and kidney ?
Reply to the general comment of the Reviewer's
In the discussion text, brief information on the determination of ROS and MDA in frozen tissues has been added (lines 484-487 highlighted in gray).
Reply to Reviewer's minor corrections:
Line 91: The lacking word “with” has been added (highlighted in grey).
line 373: The Cat activity in striatum and hippocampus was not presented in the revised version because ANOVA was insignificant. However, in order not to create the impression that these results are omitted, this information has been added to the text of the current version (lines 367-368 highlighted in grey).
Lines 391-393: Description of the post hoc analysis referring to Fig.3C has been changed according to the used Bonferroni post hoc (lines 386-388). By mistake in the revised version of manuscript, was preserved the description with the previously applied post hoc Newman-Keuls test.
Table 1 and Table 2 have been corrected.
In Fig. 7 changes in the level of bound sulfane sulfur in the liver and kidneys were not shown, because this parameter was not determined in the studied brain structures. So the results would not be complete. The determination of bound sulfane sulfur requires a large amount of tissue, therefore it is a factor limiting the possibility of performing such an assay for small amounts of tissue obtained from 16-day-old rats.

This manuscript is a resubmission of an earlier submission. The following is a list of the peer review reports and author responses from that submission.
Round 1
Reviewer 1 Report
The paper from Górny et al investigated the activity of antioxidant enzymes and ROS levels as well as the concentration of GSH, Cys and Met in tissues from a rat model of schizophrenia. The study is the sequence of a previously published paper (Górny, M. et al Molecules 2019, 24, 4253) where the effects of a GSH synthesis inhibitor (BSO) and a dopamine reuptake inhibitor (GBR 12909), administered alone or in combination, were evaluated on the levels of GSH, sulfur amino acids, DNA methylation and schizophrenia-like behavior of Sprague-Dawley rats. This additional study complements the previous in what concerns the oxidative stress status contribution to disease development.
Overall the study is well designed, clearly presented and well discussed. A few improvements are suggested to improve manuscript reading and interpretation.
1 - Although all the abbreviations are described in the abstract (first time they appear) the definition of GSH (reduced glutathione) is missing. Add in the abstract.
2 – The introduction is too long, authors should focus on the investigated problem rather than writing a review on the oxidative stress hypothesis, that is not the main investigation of this study.
3 - The description of results is too much repetitive, several paragraphs start with or contain the sentence “A two-way ANOVA …”, making the results difficult to follow. I suggest rephrasing the repeated sentences omitting the statistical method. In addition, insert a sub-section “Statistical analysis” in Methods detailing the analysis performed.
4 – The discussion is also too extensive. I agree that most of the conflicting results have to be discussed within the specific tissue, but a concise sum up of the conclusions regarding the investigated question is difficult to ascertain from this discussion. A final scheme or figure displaying the inter-related findings in the analyzed tissues would benefit interpretation of the study main conclusions.
Reviewer 2 Report
see attached file
